# AirBadminton for Physical Activity and Well-Being in Spanish Students: Post-COVID-19

**DOI:** 10.3390/ijerph20054110

**Published:** 2023-02-25

**Authors:** Mario Terol-Sanchis, María José Gomis-Gomis, Carlos Elvira-Aranda, David Cabello-Manrique, José Antonio Pérez-Turpin

**Affiliations:** 1Department of General and Specific Didactics, Faculty of Education, University of Alicante, 03690 Alicante, Spain; 2Department of Physical Education and Sports, Faculty of Sport Sciences, University of Granada, 18071 Granada, Spain

**Keywords:** AirBadminton, ports commitment, classroom climate, health

## Abstract

Sports commitment is a psychological construct that has been studied since the 1990s and that has been used in the educational field. The main objective of this study is to analyze the suitability of AirBadminton to acquired sports commitment and the classroom climate generated through the practice of AirBadminton. It was also proposed to analyze the physical, technical and temporal characteristics of AirBadminton. The research was developed with 1298 students between 13 and 15 years of age (mean ± standard deviation; body height: 1.61 ± 7.08 m; body mass 59.68 ± 7.11 kg); one group developed an AirBadminton didactic unit forming the experimental group, and a second group carried out other net games, being the control group. The following instruments were used: the Sports Commitment Questionnaire-2 CCD-2, the Brief Class Climate Scale EBCC, the analysis software LongoMatch version 1.10.1, the heart rate (HR) and the distance traveled of some participants were monitored with different Polar brand sensors (Polar H10 and Verity Sense) and two SPI-Elite GPS devices from the GPSports brand. Results show that sports commitment was increased in the experimental group. AirBadminton shows aspects that are directly and positively related to intrinsic motivation and adherence to sports practice; it improves the classroom climate and increases the desire to excel of the participants.

## 1. Introduction

We live in a period of history in which young people are passive subjects and spend much of their time in front of screens, specifically an average of 3.99 h/day [1]. We can verify that even at the end of the 1980s, 73% of the population spent their free time inactively, reading or watching TV; 23% performed physical or sports activities, such as walking, cycling or light gymnastics, and only 3% practised any sport [2]. If we go back thousands of years ago, our physical condition determined our survival, whether it was to face wild animals and thus be able to hunt, or to search for water. That kept us healthy and strong, and that is how our genes have evolved for so many years [3]. Natural selection [4] and our ability to adapt have made movement synonymous with health, so the WHO recommends for young people between 5 and 17 years of age at least 60 min of physical activity at moderate or vigorous intensity, recommending that it be aerobic in nature, and also adds that “it should be incorporated, at least three times a week, vigorous activities that strengthen, in particular, muscles and bones” [5]. According to the physical activity guide of the UK Chief Medical Officers (2019), the benefits of practising physical activity for children and young people between 5 and 18 years of age are: the improvement of confidence and social relationships, the development of coordination, increased concentration and learning, strengthening muscles and bones, improving health and fitness, maintaining a healthy weight, improving sleep and general well-being. On the other hand, its absence is one of the most determining mortality risk factors on the entire planet. According to [6], people who do not do enough physical exercise have a mortality risk between 20% and 30% higher than those who are sufficiently active. Another interesting point to know is the analysis of the motivations that exist towards the practice of physical activity. To do this, we have used the “Sports Habits Yearbook 2019”, based on the “Survey of Sports Habits in Spain 2015”.

Sports commitment is a psychological construct that has been studied since the 1990s to identify the factors that contribute to the continuous participation of athletes in sports, and that, in recent years, has been used in the educational field [7]. In Reference [8], the authors define it as “the psychological state to persist in a sport over time” (p. 2), and through their model, they try to offer a theoretical framework to examine this variable that explains why athletes continue to be involved in their sport. Factors it looks at include different types (enthusiastic and restrained engagement) and sources of engagement. Enthusiastic commitment refers to the positive aspects of commitment and involves the voluntary or “want” elements of an individual’s commitment decision. On the other hand, restricted commitment reflects a form of commitment in which athletes feel obligated or feel that they “have to” continue their participation in the sport [9]. An example of a source would be the challenges or obstacles that may arise, such as poor competitive results or the need to devote attention to other demands of life [10].

Badminton is an option to consider, since, as referred to in [11], it produces many benefits, among which we find the improvement of anatomical posture, physiological and physical capacities, and health. Various publications highlight its inclusive nature, its accessibility, its pleasant nature and state that it is a source of motivation for students [12,13]. In addition, its low learning difficulty in the initial phases allows continuity in the game, which favours effective participation in classes, motivation and adherence to its practice [14]. However, badminton is an indoor sport and is usually practised in pavilions or gyms, a feature that restricts its practice and limits opportunities. For this reason, and to provide new options, AirBadminton was born. This new modality allows creating new opportunities for all kinds of ages and, thus, being able to put it into practice on hard surfaces, on grass or sand; and not only individually, but also in pairs or trios. The purpose of the Federation is to find an effective, economic and sustainable way to motivate more and more people to do this sport. For this new outdoor modality, a new steering wheel called “AirShuttle” has been developed, a new prototype that allows you to play with winds of up to 12 km/h. AirBadminton is the adaptation that has been made to traditional badminton to allow it to be practised outdoors, ensuring continuity in the game and the good feelings of its practitioners. Different studies have resulted in its current regulations, modifying the dimensions of the court, the height of the net, the scoring system and the type of shuttlecock compared to traditional badminton [15,16]. Therefore, the aim of the study tries to answer if AirBadminton sport is capable of improving the well-being of students such as has been shown in other sports. It is also important for the classroom climate and the consequences for each adolescent.

The main objective of the study is to analyse the suitability of AirBadminton to acquired sports commitment and the classroom climate generated through the practice of AirBadminton. In addition, it was also proposed to analyse the physical, technical and temporal characteristics of AirBadminton. In this way, the first hypothesis is that the classroom climate and sports commitment will improve to a greater extent in the experimental group. The second hypothesis is that AirBadminton will be an easy sport to learn and will offer optimal physical stimulation to improve the health of secondary education students.

## 2. Materials and Methods

The research was developed with 1298 students between 13 and 15 years of age from the province of Alicante (Spain). One group developed an AirBadminton didactic unit forming the experimental group (mean ± standard deviation; body height: 1.60 ± 7.32 m; body mass 59.19 ± 6.92 kg), and a second group carried out other net games such as paddle tennis, tennis, squash and badminton, being the control group (mean ± standard deviation; body height: 1.62 ± 6.84 m; body mass 60.16 ± 6.34 kg). The inclusion criteria were regular attendance at classes and correct completion of the two questionnaires at the beginning and at the end of the intervention. Participants did not have any injury in the last 6 months and they did not take stressful or stimulant substances during the research days. In addition, they did not perform the intervention if the doctor did not recommend physical activity. Note that the participants were previously informed of their participation in the study, as well as their families and the centre’s management. Experimental procedures were approved by the local ethics committee and conducted in accordance with the Declaration of Helsinki [17].

In order to measure the sports commitment acquired by the participants and the classroom climate generated, the following instruments were used:

The Sports Commitment Questionnaire-2: SCQ-2 [9]. It is the Spanish adaptation of the Sport Commitment Questionnaire-2 (SCQ-2) [8]. The CCD-2 questionnaire is made up of a total of 58 items that reflect 12 different subscales and include 10 sources of commitment to sport, as well as two types of commitment (Enthusiastic Engagement and Restricted Engagement). The internal consistency of the different subscales obtained Cronbach’s alpha values that exceeded 0.70 in all of them except for Restricted Commitment (α = 0.62) and Social Restrictions (α = 0.68). The reliability of the instrument was a Spearman–Brown coefficient of 0.77 and a Cronbach’s alpha value of 0.87 for the two halves.

The Brief Class Climate Scale: BCCS [18]. It is a 4-point Likert-type scale made up of a total of 11 items that evaluates two dimensions of classroom climate: group cohesion and group management. The internal consistency obtained by the creators of the instrument in its validation was a Cronbach’s alpha coefficient of 0.83. Group cohesion is a dynamic horizontal dimension (student–student) that can be observed between the components of a group and refers to the degree of satisfaction, involvement and cohesion that exists between them. Group leadership is a vertical dimension (teacher–student) and refers to the way in which the teacher satisfactorily influences the development of the class through order and organisation, task orientation and the quality of their relationship with students.

To measure the physical, technical and temporal characteristics of AirBadminton, video recordings of 6 matches were made with 2 Sony HDRCX280 (Weybridge, UK) video cameras. Subsequently, the matches were analysed with the LongoMatch version 1.10.1 analysis software, and the frequency analysis was performed with Microsoft Excel 2019 spreadsheets. The recording techniques were collected by agreement between observers who have experience in the applied techniques. Likewise, the heart rate (HR) and the distance travelled of some participants were monitored with different Polar brand sensors (Polar H10 and Verity Sense) and two SPI-Elite GPS devices from the GPSports brand. GPS data was processed and analysed with the Team AMS 2.1 software, and HR data using the Polar Flow and Polar Beat apps. The physical variables analysed were the distance covered and the intra-session HR. Temporary variables refer to the duration of points, games and matches. The technical variables to the types of hits made and their effectiveness, are defined below (see Table 1).

Firstly, authorization was requested from the centre to carry out the study. Shortly thereafter, students and families were informed of the purpose of the study and informed consent was obtained from the participants. Once the instruments to be used had been prepared and the didactic intervention for both groups had been designed, the first session of the study was carried out, completing the EBCC scale on classroom climate in both groups and making video recordings of the matches and data collection with the participants. GPS and HR sensors were used with the experimental group. The sports commitment CCD-2 was completed before the second session of the group, once they had already experienced the content to work on. Approximately 8 and 25 min, respectively, were used to complete both instruments. At the end of the teaching units, which lasted a total of 12 weeks, the same procedure was followed, completing the EBCC at the beginning of the penultimate session and subsequently making the video recordings and data collection from the GPS and HR sensors. The CCD-2 was held in the last session.

Regarding the methodology used in both didactic units, it should be noted that there were no differences, both groups using traditional and cognitive teaching styles. Traditional styles were characterised by a direct instruction technique and by reproductive styles such as modified direct command or task assignment. Cognitive styles were characterised by styles such as guided discovery and by different techniques that favour inquiry [19]. Therefore, we can say that there were no methodological differences between both interventions.

The general structure of the PE sessions was as follows: initial assembly to present the contents of the session, general warm-up directed by the teacher or through activation games related to sports, the main part where different technical–tactical aspects were worked on and regulations from modified games, reduced games, mini-sports or sport as such, and final reflection.

A quasi-experimental pre-test–post-test design was adopted, with a control group [20]. Both groups received different interventions (the experimental group received AirBadminton intervention and the control group received other net games), so the independent factor or variable belonged to one or the other group, and the criterion or dependent variables were the scores of the subjects in the evaluation tests of classroom climate and sports commitment [21]. For this, the GLM (General Linear Model) of Repeated Measures was used, which analyses groups of related dependent variables that represent different measures of the same attribute [22]. To carry out all these statistical analyses, the SPSS Statistical Package, version 28.0 was used.

The analysis of the video recordings of the AirBadminton matches was carried out with the free software LongoMatch version 1.10.1. The frequency analysis was performed with Microsoft Excel spreadsheets. The program used to analyse the data from the heart rate sensors and the SPI-Elite GPS was the Team AMS v.1.5 software.

## 3. Results

Table 2 shows the means and standard deviations of the study variables at the pre-test and post-test time for the experimental and control groups. We observed how sports pleasure or enjoyment was the main predictor of sports commitment both in the experimental group (μ = 3.09 ± 1.31) and in the control group (μ = 2.95 ± 1.37). Although in the control group this source of compromise was slightly reduced (−0.10), by contrast, it was increased (+0.17) in the experimental group. It is also observed that the desire to stand out was another notable source of commitment, both in the experimental group (μ = 2.72 ± 1.44) and in the control group (μ = 2.68 ± 1.39).

Table 3 shows the results of the ANOVA for the sports commitment variables that meet the homogeneity of variances. We observe how statistically significant data were obtained in the inter-subject effects tests of all the factors of the dependent variable (*p* = 0.001). The intra-subject effects tests indicate that only the effect of the interaction between the two independent variables: time of evaluation and belonging to a group (ME-LW*GROUP) on the dependent variable, is significant in only 2 of the 12 factors, specifically in factors 11 (F = 5.540; *p* = 0.023) and 12 (F = 5.043; *p* = 0.029), both of which are related to the desire to excel. However, we must also take into account the size of the effect and the power observed, being in both cases intermediate values, so we cannot affirm categorically that the change in both factors is due to the treatment or that the treatment has been effective for improved engagement overall, but there has been a noticeable effect on both of these engagement sources.

In the case of classroom climate (Table 4 and Table 5), significant differences were obtained in the intersection of the two dimensions for the tests of between-subjects effects (*p* < 0.001) and in the tests of intra-subjects contrasts of the Group Driving dimension (F = 5.212; *p* = 0.027) and the sum of both dimensions (F = 5.331; *p* = 0.025), taking into account the differences between groups. In this sense, it should be noted that the size of the effect of the “Intra*Between” tests is notable (eta squared close to 0.1) and the resulting observed power implies that with a probability of 61%, we can say that the treatment has been effective to improve both group leadership and the classroom climate in general. However, both values are much lower than those of the “Inter” interaction test of the variables with the time of evaluation (ME) separately.

In relation to the results obtained after the analysis of the technical and temporal characteristics of AirBadminton (Table 6), after analysing 6 games and a total of 737 points, different aspects can be highlighted. In the first place, the average of the total points per game taking into account the three modalities analysed (male, female and mixed) is slightly higher in the pre-test compared to the post-test, with measurements of 63.33 points and 56.66 points, respectively. This is due to the fact that one more game was played in the pre-test and, in total, 21 more points were played compared to the post-test. However, if we look at the average number of points per game, it was very similar, with an average of 19 points in the pre-test and 18.56 in the post-test. If we look at the different modalities separately, there were no significant differences between the pre-test and posttest in the mean points per game.

In relation to the temporal variables, it should be noted that the matches lasted from 14:06 min to 21:49 min, with an average duration of the games of 5:23 min. In relation to the active playing time of the matches, they were clearly superior in the post-test, especially in the men’s match where the active time went from 2:15 to 5:27. This is due to the fact that the average duration of the points and the number of exchanges in the points was greater in the post-test in both the male and female matches, going from an average of 2.8 to 4.2 exchanges in the boys and from 2.2 to 2.7 in the girls, with points of up to 16 and 8 exchanges, respectively. That said, we can say that the continuity of the game was increased after the intervention.

Regarding the technical aspects (Table 7), the matches at the end of the intervention show a greater variety of hits and use all the technical gestures to a greater extent except the service, due to the increase in the continuity of the game. After the service, the predominant type of technical gesture in the pre-test and post-test is the Drive. However, in the post-test, the use of the Lob and the Clear is increased in both the boys’ and girls’ matches. Other changes to note are that the Smash increases significantly in the male match, and in the mixed match the use of the Drive increases. Similarly, it should be noted that 58% of the hits cause the continuity to continue at the point, compared to 18% of the pre-test. The % that a hit ends in a point is very similar (with a difference of 1%) between the pre-test and post-test, and the probability of making an error with each hit decreases by 13% in the post-test. If we analyse the effectiveness of the types of technical gestures separately, it should be noted that the drive, the lob and the service become more reliable hits in the 3 modalities, as their error percentage decreases and the percentage of continuity of the points increases.

To finish exposing the results (Table 8), we must mention the data obtained from the GPS and HR sensors that tried to analyse the physical characteristics of the sport. Regarding the HR values during the practice of AirBadminton, these ranged between 116 and 180 ppm in boys and between 114 and 166 ppm in girls, with mean HR values of 134 and 123 ppm, respectively. Furthermore, for 90% of the time they remained at values between 125 and 155 ppm. As for the distances covered and other data obtained from the GPS, it is seen that the distances covered during the PE session were similar between the different modalities and evaluation moments (pre-test and post-test), with distances covered ranging from 866 and 1458 metres per session, and maximum speeds exceeding 14 km/h in men’s and mixed matches.

## 4. Discussion

The main objective of the study is to analyse the suitability of AirBadminton on the acquired sports commitment and the classroom climate generated through the practice of AirBadminton. In addition, it was also proposed to analyse the physical, technical and temporal characteristics of AirBadminton. From the results obtained, we can say that the hypotheses raised in our study have been partially confirmed.

In relation to sports commitment, it has been seen that sports pleasure has been the main source of sports commitment in both groups, and that the AirBadminton didactic unit has served to improve two sources of commitment related to the desire to excel in order to achieve greater degree of mastery of the sport and be superior to their opponents. These results are in line with other studies, where both sources of commitment have a consistent relationship with the acquisition of enthusiastic commitment [10,23,24]. In addition, they suggest that the practice of AirBadminton could generate greater participation and enjoyment than the other net games, an aspect to be taken into account by teachers in order to try to improve the motivational regulation of students in classes [25]. Following this premise, other investigations have shown the importance of badminton at any age and gender and the benefits it brings to physical, mental and social health [26]. In addition, the importance of practising badminton can improve health in different populations, levels and pathologies [27,28].

The fact of not having obtained improvements in the rest of the sources and types of commitment may be due to different factors, such as the size of the sample, the presence of attractive alternatives [23] or the non-use of a methodology active and competent to design the didactic intervention, as suggested by [29]. However, we must emphasise that enthusiastic commitment has been predominant in the experimental group, reflecting the desire and determination to persist over time with their practice. This suggests that AirBadminton practitioners had a positive experience with this sport, and that they have not felt compelled to persist in it, an aspect that [30] links with the extrinsic aspirations of the students.

Regarding the classroom climate, it should be noted that the initial hypothesis is fulfilled, since the experimental group is the only one where the classroom climate improves. This fact could be explained by the accessible, inclusive and less difficult nature that racket and shovel sports share in comparison with other net games [13]. Or, due to the novelty of AirBadminton compared to other net games, an aspect that has been able to influence the interest, satisfaction, climate and motivation of students in PE classes [31]. The significant difference between the two groups in the group leadership dimension makes us think that the practice of AirBadminton facilitates organisational aspects and task orientation to a greater extent. According to [32,33], in the control group there may have been greater indiscipline, demotivation and conflict, aspects that would negatively affect the teacher–student socio-affective relationship [34]. These two aspects, which in [18] make up the group leadership variable, could be explained by the fact that AirBadminton is a divided-court sport that is practised in a small space, facilitating the teacher’s organisational work as far as to the distribution of students in space [35]. While other net games are practised in different spaces and facilities, this aspect may hinder the didactic intervention of the teacher and the goal orientations of the students [34].

Regarding the characteristics of AirBadminton, the results confirm the proposed hypothesis, since the students were able to improve the active times and the number of exchanges in a few sessions, and to reduce the number of errors in the most predominant technical gestures. Likewise, it should be noted that at the end of the intervention, the students still had a great margin for improvement compared to the elite game, where the effectiveness and duration of the points is greater in the three doubles modalities [36].

In relation to physical stimulation, the results obtained from moderate HR suggest that the recreational practice of AirBadminton offers an optimal stimulus for improving the physical condition and health of students, despite having obtained slightly lower values than those of other sports. In amateur female paddle HR is lower than in competitive sport [37,38]. However, the short distance achieved in the sessions where AirBadminton was exclusively played, which may have been due to the low level of physical fitness of the students [39] and/or the intrinsic characteristics of the sport, urges the teachers to use modified situations of the game that imply greater mobility [40], such as reducing the game to one-vs.-one or adding actions between hits, to obtain higher levels of motor and physiological commitment during the learning process to improve student health.

This study shows the improvement in mental and physical health after a global pandemic. The results of the study open the way for future research in the absence of these. It shows how physical activity and, more specifically, AirBadminton can be a good alternative to improve well-being parameters that are so important in today’s society. In addition, the sports commitment of the participants is interesting for future research in order to improve adherence to sport and avoid a sedentary lifestyle.

## 5. Conclusions

AirBadminton shows aspects that are directly and positively related to intrinsic motivation and adherence to sports practice. In addition, it can be a useful sports content to favour the teaching–learning process to the extent that it improves the classroom climate and increases the desire to excel of the participants. In relation to the results on the physical, technical and temporal characteristics of AirBadminton, these make us think that it is an easy sport to learn, capable of being used on several occasions and in different contexts throughout the stage, and that it offers an optimal stimulus to improve student health. Finally, the sporting commitment observed by the students is important to avoid the current sedentary lifestyle and support the practice of physical activity and sports.

## Figures and Tables

**Table 1 ijerph-20-04110-t001:** Definition of the technical variables analysed (categories and subcategories).

Category	Definition
Service	Technical gesture that starts each play from behind the service line. It is also known as the strategic complex 0 (K0)
Drive	Technical gesture made at shoulder height with a long and horizontal trajectory, either with the right or reverse face of the racket.
Clear	High hand technical gesture with a long and ascending trajectory with the intention of sending the steering wheel to the bottom of the track.
Smash	High hand technical gesture with a long and descending trajectory with the intention of sending the steering wheel to the bottom of the track.
Drop	High hand technical gesture with a horizontal and short trajectory that seeks to send the steering wheel to the rival field area closest to the network (valid).
Lob	Low-hand technical gesture with a short and ascending trajectory with the intention of sending the steering wheel to the rival field area closest to the network.
Low	Low-hand technical gesture with a short and ascending trajectory with the intention of sending the steering wheel to the rival field area closest to the network.
Active time (flyer in play)	Time, in seconds, since the service action begins until the point concludes, either because the steering wheel falls to the ground, hits the body of a player or stays on the net.
Exchanges	Number of times the steering wheel passes over the network to the opposite field with options to be played by the rival team.
Subcategory	Definition
Effectiveness or Result	Point: When the steering wheel falls directly to the rival track, when he hits the body of a rival player, when he plays the network and passes to the other field above this or when the defender touches the steering wheel with the racket in a very defective way and it falls directly to your own track or was as a result of the previous attack. Error: The blow goes out of the boundaries of the track, falls in the dead zone or impacts the network without moving to the other field, being a point for the rival. Follow: the steering wheel is hit properly, passes over the network and is played by the rival with a guarantee of success, continuing the point.

**Table 2 ijerph-20-04110-t002:** Pre-test and post-test measurements and standard deviations by group and CCD-2 factor.

CCD-2 Factor	Measure	M (DS) Experimental (N = 623)	M (DS) Control (N = 645)
Enthusiastic commitment	Before	2.22 (1.27)	2.37 (1.31)
	After	2.33 (1.27)	2.48 (1.25)
Restricted commitment	Before	1.99 (1.24)	2.27 (1.32)
	After	2.15 (1.24)	2.45 (1.25)
Sports pleasure	Before	3.09 (1.31)	2.95 (1.37)
	After	3.26 (1.25)	2.85 (1.23)
Valuable opportunities	Before	2.15 (1.28)	2.41 (1.30)
	After	2.23 (1.27)	2.42 (1.21)
Other priorities	Before	2.11 (1.28)	2.37 (1.33)
	After	2.26 (1.29)	2.33 (1.22)
Personal investments (loss)	Before	2.07 (1.21)	2,43 (1.34)
	After	2.19 (1.21)	2.42 (1.22)
Personal investments (amount)	Before	2.07 (1.24)	2.42 (1.30)
	After	2.42 (1.21)	2.46 (1.16)
Social restrictions	Before	1.91 (1.24)	2.19 (1.35)
	After	2.02 (1.27)	2.33 (1.31)
Socio-emotional support	Before	2.10 (1.30)	2.33 (1.34)
	After	2.23 (1.32)	2.37 (1.29)
Socio-instrumental support	Before	2.14 (1.32)	2.36 (1.37)
	After	2.40 (1.32)	2.40 (1.24)
Desire to excel—mastery Achievement	Before	2.48 (1.34)	2.56 (1.31)
	After	2.61 (1.33)	2.61 (1.26)
Desire to excel—social achievements	Before	2.72 (1.44)	2.68 (1.39)
	After	2.78 (1.32)	2.65 (1.26)

**Table 3 ijerph-20-04110-t003:** Summary of intra–inter subjects univariate ANOVA (assuming equal variances).

CCD-2 Factor	Source	Sum of Squares	gl	Root Mean Square	F	Next
Enthusiastic commitment	Inter	6.237	3	2.079	1.911	0.126
	Intra	906.281	833	1.088		
	Total	912.518	836			
Restricted commitment	Inter	19.540	3	6.513	6.829	0.000
	Intra	794.459	833	0.954		
	Total	814.000	836			
Sports pleasure	Inter	15.805	3	5.268	4.934	0.002
	Intra	889.445	833	1.068		
	Total	905.250	836			
Valuable opportunities	Inter	9.289	3	3.096	2.933	0.033
	Intra	879.293	833	1.056		
	Total	888.582	836			
Other priorities	Inter	8.544	3	2.848	3.077	0.027
	Intra	770.909	833	0.925		
	Total	779.453	836			
Personal investments (loss)	Inter	16.160	3	5.387	5.217	0.001
	Intra	860.119	833	1.033		
	Total	876.279	836			
Personal investments (amount)	Inter	26.104	3	8.701	8.731	0.000
	Intra	830.179	833	0.997		
	Total	856.283	836			
Social restrictions	Inter	16.933	3	5.644	5.338	0.001
	Intra	880.857	833	1.057		
	Total	897.790	836			
Socio-emotional support	Inter	8.180	3	2.727	2.287	0.077
	Intra	993.229	833	1.192		
	Total	1001.408	836			
Socio-instrumental support	Inter	13.577	3	4.526	3.969	0.008
	Intra	949.899	833	1.140		
	Total	963.475	836			
Desire to excel—mastery	Inter	3.026	3	1.009	0.959	0.412
	Intra	876.185	833	1.052		
	Total	879.211	836			
Desire to excel—social achievements	Inter	1.501	3	0.500	0.409	0.747
	Intra	1019.314	833	1.224		
	Total	1020.815	836			

**Table 4 ijerph-20-04110-t004:** Summary of intra–inter subjects univariate MANOVA.

D1. Group cohesion	**Multivariate Tests**	**gl-h**	**F**	**Next**	**η** ** ^2^ **	**Potency** ** ^a^ **
	Intra Lambda by Wilks	1.000	0.012	0.914	0.000	0.051
	ME-LW*GRUPO	1.000	2.816	0.100	0.054	0.377
	**Effects tests**	**gl**	**F**	**Sig.**	**η** ** ^2^ **	**Potency** ** ^a^ **
	Inter	1	2145.776	<0.001	0.978	1.000
	Inter error	49				

**Table 5 ijerph-20-04110-t005:** Summary of intra-subject univariate ANOVA.

Climate Dimension	Source	SC III	gl	F	Next	η^2^	**Potency** ** ^a^ **
D2. Group driving	Intra	3.389	1	0.351	0.556	0.007	0.089
	Intra* Inter	50.291	1	5.212	0.027	0.096	0.610
	Intra Error	472.788	49				
	Inter	29,873.39	1	4165.029	<0.001	0.988	1.000
	Inter error	351.449	49				
D1 + D2	Intra	2.425	1	0.099	0.755	0.002	0.061
	Intra* Inter	131.170	1	5.331	0.025	0.098	0.619
	Intra Error	1205.653	49				
	Inter	77,554.96	1	4518.656	<0.001	0.989	1.000
	Inter error	841.001	49				

**Table 6 ijerph-20-04110-t006:** Score, exchanges and average duration points, games and matches pre- and post-test.

Times and Points	2 × 2 MASC		2 × 2 FEM		2 × 2 MIX	
	Ẋ PRE	Ẋ POST	Ẋ PRE	Ẋ POST	Ẋ PRE	Ẋ POST
POINT						
Active time *	02:53	05:74	01:97	03:47	02:08	02:70
Exchanges 0–1 **	61.5%	26.7%	68.5%	63.3%	71.4%	62.5%
Exchanges 2–3 *	35.4%	56.7%	30.1%	30.6%	26.8%	32.1%
Exchanges 4–5 *	3.1%	11.7%	1.4%	4.1%	1.8%	5.4%
Exchanges +5 *	0.0%	5.0%	0.0%	2.0%	0.0%	0,0%
Int (mín-max-med)	1–7–2.8	1–16–4.2	1–7–2.2	1–8–3.7	1–5–2.5	1–6–3.9
GAME (min)	5:30	5:17	5:27	4:42	6:23	5:00
MATCH						
Active time (m)	2:15	5:27	2:23	2:50	1:44	2:39
Total time (m)	16:30	15:51	21:49	14:06	19:10	15:00
Points per game	21.33	20	18.25	17	17.66	18.56
Total points	64	60	73	51	53	56

Note. * value expressed in sec:csec; ** percentage of total exchanges.

**Table 7 ijerph-20-04110-t007:** Effectiveness and use of AirBadminton technical gestures analysed pre- and post-test.

Technical Gestures and Effectiveness	2 × 2 MASC		2 × 2 FEM		2 × 2 MIX	
	Ẋ PRE	Ẋ POST	Ẋ PRE	Ẋ POST	Ẋ PRE	Ẋ POST
SERVICE (n./%) *	67–52%	60–34%	74–58%	51–49%	56–64%	56–50%
Point	10%	10%	26%	27%	30%	23%
Error	30%	7%	32%	14%	27%	20%
Follows	60%	83%	42%	59%	43%	57%
LOB (n./%)	18–14%	29–17%	6–5%	10–10%	3–3%	6–5%
Point	0%	3%	0%	10%	33%	33%
Error	50%	21%	17%	40%	0%	33%
Follows	50%	76%	83%	50%	67%	33%
DEJ. LOW (n./%)	0–0%	4–2%	0–0%	0–0%	0–0%	0–0%
Point	0%	0%	0%	0%	0%	0%
Error	0%	25%	0%	0%	0%	0%
Follows	0%	75%	0%	0%	0%	0%
CLEAR (n./%)	7–6%	31–18%	9–7%	11–11%	10–11%	13–12%
Point	14%	6%	33%	0%	50%	8%
Error	57%	30%	56%	55%	40%	31%
Follows	29%	55%	11%	45%	10%	62%
DRIVE (n./%)	30–23%	34–19%	37–27%	31–30%	19–22%	32–28%
Point	20%	9%	21%	26%	32%	13%
Error	43%	24%	47%	26%	53%	47%
Follows	37%	68%	32%	48%	16%	41%
SMASH (n./%)	3–2%	12–7%	1–1%	0–0%	0–0%	2–2%
Point	67%	42%	100%	0%	0%	50%
Error	33%	42%	0%	0%	0%	0%
Follows	0%	17%	0%	0%	0%	50%
DEJ. HIGH (n./%)	3–2%	6–3%	3–2%	2–2%	0–0%	4–4%
Point	0%	67%	33%	50%	0%	25%
Error	100%	17%	67%	50%	0%	50%
Follows	0%	17%	0%	0%	0%	25%

Note: * Total number of hits in the match and percentage of use with respect to the total.

**Table 8 ijerph-20-04110-t008:** Data extracted from the HR bands and the pre- and post-test GPS by modality.

	Variable FC	Pretest	Posttest
2 × 2 Masc	FChalf	127 ppm	126 ppm
	FCmax	159 ppm	169 ppm
	FChalf	121 ppm	128 ppm
	FCmax	144 ppm	175 ppm
2 × 2 Fem	FChalf	114 ppm	124 ppm
	FCmax	144 ppm	157 ppm
	FChalf	123 ppm	132 ppm
	FCmax	148 ppm	164 ppm
2 × 2 Mix	FChalf ♀	137 ppm	142 ppm
	FCmax ♀	159 ppm	163 ppm
	FChalf ♂	150 ppm	134 ppm
	FCmax ♂	169 ppm	180 ppm

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
