# Peer review of "AirBadminton for Physical Activity and Well-Being in Spanish Students: Post-COVID-19"

_ijerph, 2023, doi:10.3390/ijerph20054110_

Round 1

Reviewer 1 Report

I would like to thank you for the opportunity as I feel very fortunate to be able to review this article and I would like to congratulate the authors for this work. For me, as a physical educator, this topic is very important and has a lot of value. I detail my Suggestions below and at the conclusion of my consideration.

This manuscript investigated the adequacy of the air-badminton on the sports commitment acquired and the classroom climate generated through the practice of air-badminton.

Title: the title is concrete, representative and indicative of the problem investigated in the manuscript. As a suggestion, the title should provide information about the place where the research was conducted and provide information about the group of subjects.

Summary: the summary is clear and meets the general rules for writing a good summary. However, I would like to see a better description of the sample, indicating the context. This is the most important section of the document, as it will be read many more times than even the manuscript itself, so it needs the greatest attention. A brief note on the importance of research is an excellent conclusion to a high-level summary.

introduction

As I mentioned, I find this research extremely important in contributing to the field of physical activity. I do not disagree with the justifications of the authors and I read many very good and current arguments. However, based on the stated objective, the authors are suggested to highlight the research questions that help to carry out the research and the discussion based on the findings in which the study variables appear, the study population and the expected result.

Material and method.

Instruments: it is suggested to provide more information on the short class climate scale. Original language? Was it adapted to Spanish? Was a confirmatory factor analysis performed to verify the existence of two dimensions?

Participants. This section should be better defined. I consider including a specific section called participants. This section (participants) should include the characteristics of the sample. No information on number of participants, characteristics, inclusion criteria (what is regular attendance?)- exclusion?

Statistical analysis. It is suggested that a specific section be included for this section. Information on the assessment of the assumption of normality should be included and the choice of tests should be justified according to the stated objectives. In addition, you must specify how the significance value was established.

Results: I congratulate the authors for the work done in this section. The results are easy to read for a reader who is not familiar with quantitative methodology.

Discussion: I think a great job has been done in comparing the findings with other studies. Congratulations. It is suggested that a section on practical and theoretical implications be included to assess the scope of the research.

Conclusions: they are clear and respond to the stated objectives.

Author Response

Response to Reviewer 1 Comments

Point 1: Title: the title is concrete, representative and indicative of the problem investigated in the manuscript. As a suggestion, the title should provide information about the place where the research was conducted and provide information about the group of subjects.

Response 1: AirBadminton for physical activity and well-being in Spain students: post-covid19

Point 2: Summary: the summary is clear and meets the general rules for writing a good summary. However, I would like to see a better description of the sample, indicating the context. This is the most important section of the document, as it will be read many more times than even the manuscript itself, so it needs the greatest attention. A brief note on the importance of research is an excellent conclusion to a high-level summary.

Response 2: “The research was developed with 1298 students between 13 and 15 years of age from different schools in the province of Alicante (spain)”.

sample characteristics:

Experimental group. N = 623. height: 1.60 ± 7,32m; body weight: 59,19 ± 6.92 kg; BMI: 23,12 ± 4,2 )

Control group. N = 645  height: 1.62 ± 6,84m; body weight: 60,16 ± 6,34 kg; BMI: 22,92 ± 3.8 )

Point 3: As I mentioned, I find this research extremely important in contributing to the field of physical activity. I do not disagree with the justifications of the authors and I read many very good and current arguments. However, based on the stated objective, the authors are suggested to highlight the research questions that help to carry out the research and the discussion based on the findings in which the study variables appear, the study population and the expected result.

Response 3: We agree with this point. To do this, we add several recent articles that highlight the importance of badminton in health:

(Cabello-Manrique et al., 2022); (Lassandro et al., 2021); (Liu, 2020)

Point 4: Instruments: it is suggested to provide more information on the short class climate scale. Original language? Was it adapted to Spanish? Was a confirmatory factor analysis performed to verify the existence of two dimensions?

Participants. This section should be better defined. I consider including a specific section called participants. This section (participants) should include the characteristics of the sample. No information on number of participants, characteristics, inclusion criteria (what is regular attendance?)- exclusion?

Response 4: Instruments: Is a review of 22 scales adapted to spanish by López-González & Bisquerra (2013). This scale is based on an analytical review of scientific documentation between 2000 and 2011 in different libraries and databases. reports and summaries such as the Climate Report Card and the School Climate Survey were reviewed Compendium, produced by the Education De-department of California (2011), the School Climate-te Research Summary from the Center for Social and Emotional Education of New York (2010) and Social-Emotional Learning Assessment Measures for Middle School Youth. The two dimensions were established from the review of 22 scales by a committee of experts from the Department of Research Methods and Educational Diagnosis of the University of Barcelona, Department of Social Psychology of the University of the Basque Country and 8 high school teachers.

Participants: Regular attendance refers to attending classes on a regular basis. This refers to 90% of the classes.

Exclusion criteria: Not having had a serious injury in the last 6 months. Do not take stressful or stimulating substances during the research days. Do not perform the intervention if the doctor does not recommend physical activity.

López-González, L.; Bisquerra, R. Validación y análisis de una escala breve para evaluar el clima de clase en Educación Secundaria. Isep Science, 2013, 5, 62-77.

Point 5: Statistical analysis. It is suggested that a specific section be included for this section. Information on the assessment of the assumption of normality should be included and the choice of tests should be justified according to the stated objectives. In addition, you must specify how the significance value was established.

Response 5: We consider that the ANOVA test is specific for this type of study because it allows us to compare the variance of the means of the different groups. We believe that the ANOVA clearly shows the differences between the groups and the answers obtained in the questionnaires due to the sample and its characteristics. Thank you.

Point 6: It is suggested that a section on practical and theoretical implications be included to assess the scope of the research.

This study shows the improvement in mental and physical health after a global pandemic. The results of the study open the way for future research in the absence of these. It shows how physical activity, and more specifically Airbadminton, can be a good alternative to improve well-being parameters that are so important in today's society. In addition, the observed sports commitment of the participants is interesting in future research in order to improve adherence to sport and avoid a sedentary lifestyle.

 Thank you!

bibliography corresponding to point 3:

Cabello-Manrique, D., Lorente, J. A., Padial-Ruz, R., & Puga-González, E. (2022). Play Badminton Forever: A Systematic Review of Health Benefits. International Journal of Environmental Research and Public Health, 19(15). https://doi.org/10.3390/ijerph19159077

Lassandro, G., Trisciuzzi, R., Palladino, V., Carriero, F., Giannico, O. V., Tafuri, S., Valente, R., Gianfelici, A., Accettura, D., & Giordano, P. (2021). Psychophysical health and perception of well-being between master badminton athletes and the adult italian population. Acta Biomedica, 92(4). https://doi.org/10.23750/abm.v92i4.9857

Liu, J. (2020). Discussion on Physical Training for Badminton in Colleges and Universities Based on Fitness Purpose. 416(Iccese), 831–834. https://doi.org/10.2991/assehr.k.200316.182

Reviewer 2 Report

Dear Authors,

First of all, I would like to congratulate the authors for their work and the novelty in the practice of this new sports modality or sports alternative.

Next, I proceed to make several notes that must be corrected to improve the text of your investigation.

Line 34  Close quotes

There are sentences that are included or that do not start, it is understood that it is due to the way of quoting from the magazine, but some twist must be made to make the sentence make sense. As is the case of Line 52 with reference [9], it also happens in line 45 with reference [7]. All text must be reviewed.

Line 65  Capital letter in “various”

A uniformity must be maintained with the name Air Badminton, sometimes it is presented together and other times it appears separately, review the entire document.

It is recommended to refer to the ethics committee of the university of origin that has given its approval for this research or to make it clear that the Declaration of Helsinki is complied with (WMA 2021, Bošnjak 2001, Tyebkhan 2003), which establishes the fundamental ethical principles. for research with humans.

It is recommended to list the types of network sports that have been applied to the control group, since Ping-Pong is not the same as tennis, or paddle tennis or badminton itself.

Considering the technical analysis of airbadminton, the methodology used for this second study should be pointed out, which deepens the first part of the investigation.

In reference to the observation instrument:

It is recommended to state how the instrument was created, if it is for specific use, if it has a name, the place where these variables are obtained (committee of experts, federative manuals...).

In relation to the technical record, it should be said who has done it? (observer requirements), if it has been specifically trained, agreement between observers, etc….

Considering the results section, it should be improved, since they are mixed with the discussion. The results must be aseptic, do not show interpretations. Your new wording is recommended.

The conclusions are very general and more can be provided with the data obtained.

Finally, at a general level, I must say that the inclusion of the technical analysis exclusively of Air badminton is not understood, it is considered to have great potential for a different investigation, I recommend excluding it from the document, or integrating it more appropriately.

Author Response

Response to Reviewer 2 Comments

Point 1: Line 34  Close quotes

There are sentences that are included or that do not start, it is understood that it is due to the way of quoting from the magazine, but some twist must be made to make the sentence make sense. As is the case of Line 52 with reference [9], it also happens in line 45 with reference [7]. All text must be reviewed.

Response 1: We have reviewed all the references and we consider that they are in accordance with the statutes of the journal. Thank you!

Line 65  Capital letter in “various”. Okey. Thank you.

A uniformity must be maintained with the name Air Badminton, sometimes it is presented together and other times it appears separately, review the entire document.

Airbadminton placed throughout the article. Thank you.

It is recommended to refer to the ethics committee of the university of origin that has given its approval for this research or to make it clear that the Declaration of Helsinki is complied with (WMA 2021, Bošnjak 2001, Tyebkhan 2003), which establishes the fundamental ethical principles. for research with humans.

Thank you very much for that!

According to WMA, the last valid declaration found would be:

(Holstila et al., 2013)

Holstila, E., Vallittu, A., Ranto, S., Lahti, T., & Manninen, A. (2013). World Medical Association Declaration of Helsinki. JAMA, 310(20), 2191. https://doi.org/10.1001/jama.2013.281053

It is recommended to list the types of network sports that have been applied to the control group, since Ping-Pong is not the same as tennis, or paddle tennis or badminton itself.

The control group plays paddle tennis, tennis, squash, fronton and badminton. Thank you.

Considering the technical analysis of airbadminton, the methodology used for this second study should be pointed out, which deepens the first part of the investigation.

In reference to the observation instrument:

It is recommended to state how the instrument was created, if it is for specific use, if it has a name, the place where these variables are obtained (committee of experts, federative manuals...).

Airbadminton was created to reinvent badminton. The international badminton federation is interested in this project to give greater visibility to this sport. It is a future project whose rules are already established as well as the use of material, dimensions and type of game. It is not a regulated sport yet.

In relation to the technical record, it should be said who has done it? (observer requirements), if it has been specifically trained, agreement between observers, etc….

This is a good suggestion. The observers have experience in this sport. They have been participating in this project of the international federation for years. In the case of the study, there is an agreement between observers.

The conclusions are very general and more can be provided with the data obtained.

Thank very much. Yes, We believe it is important to add the importance of adherence to sport due to the responses observed on sports commitment.

Reviewer 3 Report

I propose to expand the discussio, the work will be more valuable. This is a more recent reference to literature.

Author Response

Response to Reviewer 3 Comments

Point 1 I propose to expand the discussio, the work will be more valuable. This is a more recent reference to literature.

Response 1: Thanks for your proposal. We are considering adding more recent bibliography to expand the article.

Cabello-Manrique, D., Lorente, J. A., Padial-Ruz, R., & Puga-González, E. (2022). Play Badminton Forever: A Systematic Review of Health Benefits. International Journal of Environmental Research and Public Health, 19(15). https://doi.org/10.3390/ijerph19159077

Lassandro, G., Trisciuzzi, R., Palladino, V., Carriero, F., Giannico, O. V., Tafuri, S., Valente, R., Gianfelici, A., Accettura, D., & Giordano, P. (2021). Psychophysical health and perception of well-being between master badminton athletes and the adult italian population. Acta Biomedica, 92(4). https://doi.org/10.23750/abm.v92i4.9857

Liu, J. (2020). Discussion on Physical Training for Badminton in Colleges and Universities Based on Fitness Purpose. 416(Iccese), 831–834. https://doi.org/10.2991/assehr.k.200316.182

Chen, C. C., Ryuh, Y. J., Donald, M., & Rayner, M. (2022). The impact of badminton lessons on health and wellness of young adults with intellectual disabilities: a pilot study. International Journal of Developmental Disabilities, 68(5), 703–711. https://doi.org/10.1080/20473869.2021.1882716

Mohammadi, M. (2011). A study and comparison of the effect of team sports (soccer and volleyball) and individual sports (table tennis and badminton) on depression among high school students. Australian Journal of Basic and Applied Sciences, 5(12), 1005–1011.

Reviewer 4 Report

Strengths:

1. The topic of the article is an interesting one, considering the fact that it refers to the involvement of students in sports games (air badminton). We also appreciate the interest and concern for promoting outdoor games among students.

Questions/Recommendations for Authors:

1. Line 27 refers to an article published in 2014. Since then, technology has evolved and the time children and young people spend in front of screens has changed. It would be advisable to refer to a more recent source as well.

2. We appreciate the large number of participants in the experiment. When was the experiment conducted? During physical education classes? In non-formal activities?

3. Is air badminton included in the school curriculum, in the category of sports games, or was it proposed only for experiment?

Author Response

Response to Reviewer 4 Comments

Point 1 . Line 27 refers to an article published in 2014. Since then, technology has evolved and the time children and young people spend in front of screens has changed. It would be advisable to refer to a more recent source as well.

Response 1: We fully agree with your contribution. We are going to add a much more recent article in which more time is shown in front of screens.

(Nagata et al., 2022)

Nagata, J. M., Ganson, K. T., Iyer, P., Chu, J., Baker, F. C., Pettee Gabriel, K., Garber, A. K., Murray, S. B., & Bibbins-Domingo, K. (2022). Sociodemographic Correlates of Contemporary Screen Time Use among 9- and 10-Year-Old Children. Journal of Pediatrics, 240, 213-220.e2. https://doi.org/10.1016/j.jpeds.2021.08.077

Point 2 . We appreciate the large number of participants in the experiment. When was the experiment conducted? During physical education classes? In non-formal activities?

Response 2: Yes, they have been done during physical education hours.

Point 3 . Is air badminton included in the school curriculum, in the category of sports games, or was it proposed only for experiment?

Response 3: It is a project that has been carried out for years with the international badminton federation. For now, it's an experiment

Round 2

Reviewer 1 Report

I would like to thank the authors for their efforts to improve the manuscript and for taking my contributions into account. I consider that this article could be published.